

# Field-obtained carbon and nitrogen uptake rates of phytoplankton

# in the Laptev and East Siberian seas

**Sang Heon Lee[1\*], Jang Han Lee[1], Howon Lee[1], Jae Joong Kang[1], Jae Hyung Lee[1],**

**Dabin Lee[1], SoHyun An[1], Dean A. Stockwell[2], Terry E. Whitledge[2]**

\* corresponding author

[1]Department of Oceanography, Pusan National University, Busan 609-735, Korea

[2]Institute of Marine Science, University of Alaska, Fairbanks, AK 99775, USA



## Abstract

The Laptev and East Siberian seas are the least biologically studied region in the Arctic Ocean, although they are highly dynamic in terms of active processing of organic matter impacting the transport to the deep Arctic Ocean. Field-measured carbon and nitrogen uptake rates of phytoplankton were conducted in the Laptev and East Siberian seas as part of the NABOS (Nansen and Amundsen Basins Observational System) program. Major inorganic nutrients were mostly depleted at 100-50% light depths but were not depleted within the euphotic depths in the Laptev and East Siberian seas. The water column-integrated chl-$a$ concentration in this study was significantly higher than that in the western Arctic Ocean (t-test, p > 0.01). Unexpectedly, the daily carbon and nitrogen uptake rates in this study (average ± S.D. = 110.3 ± 88.3 mg C m$^{-2}$ d$^{-1}$ and 37.0 ± 25.8 mg N m$^{-2}$ d$^{-1}$, respectively) are within previously reported ranges. Surprisingly, the annual primary production (13.2 g C m$^{-2}$) measured in the field during the vegetative season is approximately one order of magnitude lower than the primary production reported from a satellite–based estimation. Further validation using field-measured observations is necessary for a better projection of the ecosystem in the Laptev and East Siberian seas responding to ongoing climate change.

## Keywords

Laptev and East Siberian seas, NABOS, carbon and nitrogen uptake rates of phytoplankton



## 1. Introduction

The most dramatic environmental change in the Arctic Ocean has been the rapid and extensive decrease in sea ice extent and thickness over the recent decades (Comiso 2006; Overland and Wang, 2013; Overland et al., 2014). Sea ice is a major controlling factor for primary production of pelagic phytoplankton by modulating water column stratification and light fields (Hill et al. 2005; Gradinger 2009; Bélanger et al., 2013), although nutrient supply to the surface water have been proposed as the main controlling factor in seasonally ice-free open waters (Tremblay and Gagnon, 2009). Consequently, these sea ice changes will affect primary productivity as well as physiological status of primary producers (Lee et al. 2008, 2010; more refs) and thus carbon cycling in the Arctic Ocean (Arrigo et al. 2008; Bates and Mathis 2009; Cai et al. 2010). Some evidence for the impacts of environmental changes on phytoplankton have been already reported in various regions in the Arctic Ocean (Arrigo et al., 2008; Li et al., 2009; Wassmann et al., 2011; Ardyna et al., 2014). Several studies have reported increasing signs of annual primary production due to enhanced light availability to phytoplankton as a main consequence of recent increasing open area and longer open period in the Pan-Arctic regions from 1998 to 2009 (Arrigo et al. 2008; Arrigo and Dijken, 2011). In contrast, a restrained primary production was reported as a result of increasing cloudiness in the Arctic Ocean (Bélanger et al., 2013) due to warmer temperature and moisture fluxes in



newly open waters during summer and early fall (Eastman and Warren, 2010; Vavrus et al.,
2010). To date increasing or decreasing in the primary production of primary producers as a
consequence of ongoing environmental changes is still being debated in the Arctic Ocean
(Lee and Whitledge, 2005; Coupel et al., 2015). However, it is clear that these
environmental changes will have great effects on the ecosystem from altering the patterns of
primary production to changing the trophic structure and the elemental cycling pathways
(Grebmeier et al. 2006).
The Laptev and East Siberian seas are situated on the widest and shallowest continental
shelf in the world. Both seas are highly dynamic in terms of organic matter production and
processing, impacting the atmospheric exchange and the transport of organic matter to the
deep Arctic Ocean (Semiletov et al., 2005; Anderson et al., 2009). However, the Laptev and
East Siberian seas are among the least biologically studied regions in the Arctic Ocean
(Semiletov et al., 2005; Arrigo and Dijken, 2011). Although various physical data on
hydrography and ocean circulation have been reported by the continuous NABOS (Nansen
and Amundsen Basins Observational System) program (Bauch et al., 2014; Aksenov et al.,
2011; Polyakov et al., 2007; Dmitrenko et al., 2006), no *in situ* measurements of recent
phytoplankton productivity or nutrient concentrations have been conducted in the Laptev or
East Siberian seas during the program.



In this study, *in situ* carbon and nitrogen uptake rates of phytoplankton were measured to
quantify the primary productivity and evaluate nitrogen uptake in the Laptev and East
Siberian seas as part of the NABOS program. These data will provide the basic groundwork
for future monitoring of the marine ecosystem as it responds to ongoing climate change in
the Laptev and East Siberian seas and will provide valuable *in situ* measurements for
validating the ranges of phytoplankton primary production estimated from satellite ocean
color data.

**2. Materials and Methods**
Field-measured carbon and nitrogen uptake rates of phytoplankton were measured at
19 monitoring stations selected from a total of 116 NABOS stations (Fig. 1; Table 1) in the
Laptev and East Siberian seas from August 21 to September 22, 2013 onboard the Russian
vessel *"Akademik Fedorov"*. After samples for concentrations of major inorganic nutrient
and chlorophyll-*a* (chl-*a*) were collected at 19 productivity stations, they were analyzed
onboard during the cruise. Nutrient concentrations (nitrate, nitrite, ammonia, phosphate, and
silicate) were analyzed using an Alpkem Model 300 Rapid Flow Nutrient Analyzer (5
channels) based on the method of Whitledge et al. (1981). Total and size-fractionated chl-*a*
samples were obtained from 6 light depths (100, 50, 30, 12, 5, and 1 %) and 3 light depths



(100, 30, and 1%), respectively. The chl-*a* samples were prepared based on the same
procedure reported from previous studies in the Arctic Ocean (Lee et al., 2005; Lee et al.,
2012). Water samples for chl-*a* concentrations were filtered onto Whatman GF/F (24 mm)
and samples for size-fractionated chl-*a* were passed sequentially through 20 μm and 5 μm
pore-sized Nucleopore filters (47 mm) and 0.7 μm pore-sized Whatman GF/F filters (47
mm). The filters were kept frozen in a freezer (-80 °C) before further analysis. The frozen
chl-*a* samples were extracted in 90% acetone at −5°C for 24 hours, and the concentrations
were measured on board using a pre-calibrated Turner Designs model 10-AU fluorometer.
On-deck incubations for carbon and nitrogen uptake rates of phytoplankton were conducted
using a $^{13}C$-$^{15}N$-dual tracer technique previously performed in various regions of the Arctic
Ocean (Lee and Whitledge 2005; Lee et al., 2007 & 2012, Yun et al., 2015). Six *in situ*
photic depths (100, 50, 30, 12, 5, and 1%) were determined at each station by converting
Secchi disc depth to light intensity. Seawater samples at each light depth were transferred
from the Niskin bottles to acid-cleaned polycarbonate incubation bottles (approximately 1 L)
matched each light depth. Then, heavy isotope-enriched (98−99 %) solutions of $H^{13}CO_3$,
$K^{15}NO_3$, or $^{15}NH_4Cl$ were added to the polycarbonate incubation bottles at concentrations of
~0.3 mM, ~0.8 μM, and ~0.1 μM for $^{13}CO_2$, $^{15}NO_3$, and $^{15}NH_4$, respectively. The carbon
isotope enrichment was 5−10% of the total inorganic carbon in the ambient water



determined during the cruise. In contrast, the concentrations of $^{15}NO_3$ and $^{15}NH_4$ additions
were greater than 10 % of the ambient nitrate and ammonium concentrations at several
stations with very low concentrations. After 4 to 6 hour incubations on deck, the filters used
for the isotopic measurements as well as particulate organic carbon (POC) and nitrogen
(PON) were immediately preserved at –20°C for further mass spectrometric analysis
(Finnigan Delta+XL) in the stable isotope laboratory of University of Alaska Fairbanks, US.
The uncertainties for $\delta^{13}C$ and $\delta^{15}N$ measurements were ± 0.1‰ and ± 0.3 ‰, respectively.
Calculations of the carbon and nitrogen uptake rates of phytoplankton were based on the
methods from Hama et al. (1983) and Dugdale and Goering (1967). Carbon uptake rates
were obtained as follows:
Carbon uptake rate = $POC_{incubation} \times [^{13}C_{excess}/ (^{13}C_{enriched} * t)]$,
where $POC_{incubation}$ is the concentration of particulate organic carbon after incubation,
$^{13}C_{excess}$ is the excess $^{13}C$ [concentration of $^{13}C$ in the particulate phase after incubation −
natural abundance of $^{13}C$ in the particulate phase], $^{13}C_{enriched}$ is the $^{13}C$ enrichment in the
dissolved fraction, and t is the time duration of incubation in hours. Nitrogen uptake rate was
obtained same as carbon uptake rate. Dark carbon uptake values were subtracted from light
carbon uptake values since the measured dark uptake rates were assumed from bacterial
processes (Gosselin et al. 1997). Integrated values of the carbon and nitrogen uptake rates of



phytoplankton were calculated from surface (100 %) to 1 % light depths based on the
trapezoidal rule. The *f* ratio was calculated as a fraction of nitrate uptake rate to the sum of
nitrate and ammonium uptake rates in this study (Eppley and Peterson, 1979).


**3. Results and Discussion**
The sea surface temperature and salinity ranged from −1.76 °C to 1.62 °C and 28.29 to
33.44, respectively (Table 1). Sea ice concentration averaged during the cruise period in
2013 retrieved from National Snow & Ice Data Center ranged from 0 % to 100 % (Table 1).
The concentrations of major inorganic nutrients (nitrite+nitrate, ammonium, phosphate, and
silicate) were integrated from surface to 50 m water depth because the average euphotic
water column was 49.6 m (S.D. = ± 10.6 m) during our cruise period in 2013 (Fig. 2a-d).
The concentrations of nitrite+nitrate and ammonium were 19.3-189.3 mmol m$^{-2}$ and 2.5-39.7
mmol m$^{-2}$, respectively (Fig. 2a & b). The concentration of DIN (nitrite+nitrate+ammonium)
was 25.8-213.7 mmol m$^{-2}$. The concentrations of phosphate and silicate were 7.6-39.7 mmol
m$^{-2}$ and 19.5-329.7 mmol m$^{-2}$, respectively (Fig. 2c & d). Generally, high concentrations of
nitrite+nitrate and phosphate were found at AF005, AF068, AF071, and AF100 and they
were relatively higher in the Laptev Sea than in the East Siberian Sea (Fig. 2a & c). In



contrast, the pattern of silicate concentration is appeared opposite as those of nitrite+nitrate
and phosphate. The silicate concentration was higher in the East Siberian Sea than in the
Laptev Sea (Fig. 2d). Generally, the integrated nutrient concentrations were not depleted
within the euphotic depths. Rather, they (except silicate) were nearly depleted in the upper
layers (< 10 m), which represented approximately 50 % light depth in this study. Our
findings are consistent with the previous results in the Laptev and East Siberian seas
obtained by Codispoti and Richards (1968) who observed that the concentrations of
phosphate and nitrate are so low as to indicate nutrient limitation for phytoplankton
production in the upper layers. However, our stations were substantially deeper (> 200 m
bottom depth; Table 1) than those in Codispoti and Richards (1968) which were generally
located in shallow shelf regions (< 50 m).
Water column-integrated chl-$a$ concentration from surface to 50 m water depth
ranged from 9.9 mg chl-$a$ m$^{-2}$ at AF036 to 59.8 mg chl-$a$ m$^{-2}$ at AF091 (average ± S.D. =
25.7 ± 14.2 mg chl-$a$ m$^{-2}$) in this study (Fig. 3). In comparison to other deep waters in the
western Arctic Ocean, this range is significantly higher (t-test, $p > 0.01$) than those in the
ice-free deep waters (6.4-24.8 mg chl-$a$ m$^{-2}$) and the newly opened deep waters (7.1-15.1 mg
chl-$a$ m$^{-2}$) in the northern Chukchi Sea from mid-August to early September in 2008 (Lee et
al., 2012) and in the Canada Basin from mid-August to early September in 202 (1.6-16.7 mg





chl-$a$ m$^{-2}$; Lee and Whitledge, 2005). Furthermore, the recent study in the northeast Chukchi
Sea and the western Canada Basin showed a similar lower range of euphotic-ingrated chl-$a$
concentration (8.3-9.7 mg chl-$a$ m$^{-2}$) during the early summer period with mostly sea ice
cover from mid-July to mid-August, 2010 (Yun et al., 2015). Generally, small sized-cells
(0.7-5 μm) of phytoplankton appear to be dominant in the euphotic water columns of the
Laptev and East Siberian seas based on the size-fractionated chl-$a$ data in this study (Fig. 4).
The contributions of small sized-cells averaged from all the stations were 63.3 (S.D. = ±
17.5 %), 61.4 (S.D. = ± 19.9 %), and 59.0 % (S.D. = ± 18.4 %) at 100, 30, and 1 % light
depths, respectively. These ranges are similar to that (64.3 %) in the western Canada Basin
(Yun et al., 2015). However, the contributions of small-sized cells in the Canada Basin
reported by Lee and Whitledge (2005) are somewhat higher (69.3 %) at surface but lower
(44.4 %) at chl-a maximum layer at 50-60 m than those in this study although they were not
statistically significant (t-test, $p > 0.05$).

The water column-integrated hourly carbon uptake rate was 0.89-16.60 mg C m$^{-2}$ h$^{-1}$

(average ± S.D. = 4.83 ± 3.52 mg C m$^{-2}$ h$^{-1}$) in this study (Fig. 5). The highest rate was
observed at AF019 and the lowest rate was at AF005. The remarkably high uptake rate at
AF019 among other productivity stations were mainly due to relatively higher particulate
organic carbon (POC) concentrations and specific carbon uptake rates at upper light depths





(> 30 % light level). Vertically, the hourly carbon uptake rates were generally highest at the
100-50% light levels among the six different light depths (data not shown). The water
column-integrated hourly nitrate and ammonium uptake rate ranges were 0.05-1.96 mg N m$^{-}$
$^2$ h$^{-1}$ (average ± S.D. = 0.48 ± 0.52 mg N m$^{-2}$ h$^{-1}$) and 0.19-3.37 mg N m$^{-2}$ h$^{-1}$ (average ± S.D.
= 1.06 ± 0.76 mg N m$^{-2}$ h$^{-1}$), respectively (Fig. 6). Generally, the ammonium uptake rates
were relatively higher than the nitrate uptake rates during our cruise period. The total
nitrogen (nitrate + ammonium) uptake rate ranged from 0.25 mg N m$^{-2}$ h$^{-1}$ at AF044 to 4.49
mg N m$^{-2}$ h$^{-1}$ at AF019. No specific pattern was observed in the spatial distribution of the
carbon and nitrogen uptake rates of phytoplankton in this study.
Assuming a 24-h photoperiod per day during the summer period in the Arctic Ocean
(Subba Rao and Platt 1984; Lee and Whitledge 2005; Lee et al., 2010), the daily carbon and
nitrogen (nitrate + ammonium) uptake rates of phytoplankton varied substantially, with
ranges of 9.9-398.3 mg C m$^{-2}$ d$^{-1}$ (average ± S.D. = 110.3 ± 88.3 mg C m$^{-2}$ d$^{-1}$) and 6.0-107.7
mg N m$^{-2}$ d$^{-1}$ (average ± S.D. = 37.0 ± 25.8 mg N m$^{-2}$ d$^{-1}$), respectively. Although the water
column-integrated chl-*a* concentration is significantly higher (approximately five-fold) in
this study than in other deep waters in the western Arctic Ocean, as previously mentioned
above, our mean daily carbon uptake rate (110.9 mg C m$^{-2}$ d$^{-1}$) are relatively equivalent to
previously reported rates (Cota et al., 1996; Lee and Whitledge, 2005). Cota et al. (1996)





and Lee and Whitledge (2005) obtained rate of 123.5 mg C m$^{-2}$ d$^{-1}$ and 106 mg C m$^{-2}$ d$^{-1}$ in
the Canada Basin, respectively. However, the mean daily carbon rates in the northeast
Chukchi Sea (29.8 mg C m$^{-2}$ d$^{-1}$) and western Canada Basin (20.6 mg C m$^{-2}$ d$^{-1}$) reported by
Yun et al. (2015) were substantially lower than those in other measurements. These lower
values are mainly due to their measurements conducted in the early ice-opening season with
a considerably heavy sea ice concentration over 70 % (Yun et al., 2015). The sea ice
concentration in this study ranged widely from 0 % to 100 %, mostly ice free conditions
with an average of 20 % during the cruise period (Table 1).

The mean daily nitrogen uptake rate (average ± S.D. = 37.0 ± 25.8 mg N m$^{-2}$ d$^{-1}$) is

comparable to the rates previously reported in the western Arctic Ocean (Lee and Whitledge,
2005; Lee et al., 2012). Lee and Whitledge (2005) measured 30.5 mg N m$^{-2}$ d$^{-1}$ (S.D. = ±
16.2 mg N m$^{-2}$ d$^{-1}$) in the Canada Basin, and Lee et al. (2012) observed 33.4 mg N m$^{-2}$ d$^{-1}$
(S.D. = ± 18.4 mg N m$^{-2}$ d$^{-1}$) in the deep, ice-free northern Chukchi Sea. Based on the nitrate
and ammonium uptakes rates, the *f*-ratio averaged from all the productivity stations in this
study was 0.28 (S.D. = ± 0.17; Fig. 7), which is comparable to the average *f*-ratios (0.22-
0.34) previously reported in the western Arctic Ocean (Lee and Whitledge, 2005; Lee et al.,
2012; Yun et al., 2012; Codispoti et al., 2013). The low *f*-ratio estimated in this study
indicates that the predominant nitrogen source for phytoplankton growth was ammonium at





that time of sampling. There are two possible explanations for that result: the low amount of
nitrate available for phytoplankton growth (Kim et al., 2015) and the existence of low light
growth conditions (Lee and Whitledge, 2005; Yun et al., 2012). The nitrate uptakes of
phytoplankton are reported to be more strongly coupled with light than ammonium uptakes
(Dortch and Postel 1981). Relatively low $f$-ratios were observed overall in the Laptev and
East Siberian seas even though there were some nitrate concentrations available within the
euphotic water depths. In fact, no strong relationship between $f$-ratio and euphotic water
depth-integrated concentration of nitrite+nitrate was found in this study ($R^2$ = 0.02). This
provides indirect evidence for potential light-limited conditions for phytoplankton growth in
the Laptev and East Siberian seas during the study period.

Codispoti et al. (2013) suggests that a nutrient-limited condition of phytoplankton

production exists in the Laptev and East Siberian seas because of limited inorganic nutrient
availability (phosphate and nitrate) at the surface (Codispoti and Richards, 1968). Generally,
carbon/nitrogen (C/N) ratios have been used to an indicator of nutrient condition of
phytoplankton (Smith and Sakshaug 1990; Lee and Whitledge, 2005; Yun et al., 2012). For
example, high C/N ratios are often indicative of nitrogen deficiency for phytoplankton
growth (Smith and Sakshaug 1990). The C/N ratio of particulate organic matters and
assimilated C/N ratio averaged from all the productivity stations were 7.23 (± 5.51) and 7.03





(± 5.14), respectively in this study. These C/N ratios similar to the Redfield ratio (6.6)
indicate no strong nutrient-limited condition of phytoplankton production during the cruise
period in 2013. In this study, the relatively lower daily carbon uptake rate despite of
significantly higher chl-*a* concentration could be caused by a potential light-limited growth
condition of phytoplankton in the Laptev and East Siberian seas. In fact, there was no strong
relationship between water column-integrated POC and chl-*a* concentrations as
representative biomass for phytoplankton and daily carbon uptake rate in this study ($R^2$ =
0.001), which indicates that phytoplankton biomass itself is not a main factor for the
different carbon uptake rates from the productivity stations during this cruise period.
However, no relationship between POC and chl-*a* concentrations in this study could be
caused by natural characteristics of POC samples since it normally includes all suspended
organic carbon (detritus, bacteria, microzooplankton, etc.) as well as phytoplankton carbon.
In general, the ratio of C/chl-*a* is lower for phytoplankton under low light conditions than
under high light conditions, although it is highly variable depending on all other
environmental conditions that affect the growth rate of phytoplankton (Smith and Sakshaug,
1990). Based on cultures of polar or subpolar phytoplankton, the ratio ranges from 20-50 in
low light conditions to 100-200 in high light conditions (Smith and Sakshaug, 1990). In fact,
Lee and Whitledge (2005) found that ratios in the Canada Basin (16.8 at approximately 2%



249 light level of surface irradiance to 314 at surface water) were comparable to the laboratory-

250 measured ratios. In this study, the C/chl-*a* ratio averaged from all the productivity stations

251 was 290.8 (S.D. = ± 164.4), which indicates no light-limited condition. Since POC contains

252 not only phytoplankton carbon but also all suspended organic carbon, the C/chl-*a* ratio might

253 not be a good indicator in the Laptev and East Siberian seas with large terrestrial inputs

254 during the ice-free summer season (Macdonald et al., 2010; Anderson et al., 2011). At this

255 point, we do not have a good explanation for the lower carbon uptake rate despite of

256 substantially high chl-*a* concentration of phytoplankton in the Laptev and East Siberian seas

257 during our observation period. Based on various indicators, the growth of phytoplankton in

258 the Laptev and East Siberian seas may have experienced a light-limited condition as well as

259 the nutrient-limited condition, both of which are generally considered to occur in the Arctic

260 Ocean during the summer periods (Codispoti and Richards, 1968; Codispoti et al., 2013).

261  Assuming a 120-day growing season and the same daily productivity over the

262 season in the Arctic Ocean (Gosselin et al. 1997; Lee and Whitledge 2005; Lee et al. 2012),

263 the annual primary production (13.2 g C m$^{-2}$) during the arctic vegetative season estimated

264 in this study is comparable to the indirect estimation (9.6 g C m$^{-2}$) from drawdown

265 measurements of dissolved inorganic carbon in the East Siberian Sea (Anderson et al.,

266 2011). However, these productions are substantially (approximately one order magnitude)





lower than the mean productions (101-121 g C m$^{-2}$) in the Laptev and Siberian seas for
1998-2009 estimated from satellite-based measurements (Arrigo and Dijken 2011). An
overestimation of satellite-derived primary production in the Arctic Ocean is generally
caused by an overestimation of chl-$a$ concentration from massive colored dissolved organic
matter (CDOM) of terrestrial origin and degraded phytoplankton (Guéguen et al., 2007;
Matsuoka et al., 2011). Indeed, large terrestrial inputs of dissolved and particulate organic
matter are transported from substantial inputs of river runoff such as from the Lena,
Indigirka, and Kolyma rivers to the shelves of the Laptev and East Siberian seas during the
ice-free summer season (Macdonald et al., 2010; Anderson et al., 2011). Arrigo and Dijken
(2011) also discussed a potential overestimation by the CDOM which causes some
overestimation in surface chl-$a$ and thus net primary production from satellite-based
approaches. However, they argued that the overestimation of net primary production as high
as 6.1 % is nearly equivalent to the underestimated portion (7.6 %) by missing subsurface
chl-$a$ maximum (SCM) layer in the Arctic Ocean. In this study, the SCM layers were
detected but not common in overall productivity stations (6 stations out of 19 productivity
stations) during the cruise period (data not shown). However, our measured annual
production is surprisingly lower compared to the satellite-derived production in the Laptev
and East Siberian seas, although our productivity in this study were executed at one time



period in 2013. Further careful validation between the two different methods (field
measurement vs. satellite-derived approach) is needed for a better understanding of the least
biologically studied region undergoing severe and ongoing environmental changes in the
Arctic Ocean.

**4. Summary and Conclusion**

Field-measured phytoplankton productivity and nutrient concentrations were obtained in

the Laptev and East Siberian seas, one of the least biologically studied regions in the Arctic
Ocean (Semiletov et al., 2005; Arrigo and Dijken, 2011), during the NABOS (Nansen and
Amundsen Basins Observational System) cruise from August 21 to September 22, 2013
(Fig. 1).

During the cruise period, the nutrient concentrations within the euphotic depths were not

depleted although they were depleted in the upper layers which are consistent with the
previous results (Fig. 2). The euphotic water column-integrated chl-*a* concentration (25.7 ±
14.2 mg chl-*a* m$^{-2}$; Fig. 3) was significantly higher in this study than those previously
reported in the other parts of the Arctic Ocean (Lee and Whitledge, 2005; Lee et al., 2012).
Among the different cell sizes of phytoplankton, small phytoplankton were dominant
(approximately 60 %) in the Laptev and East Siberian seas (Fig. 4). Based on the low *f*-ratio



(0.28 ± 0.17; Fig. 7) observed in this study, ammonium appears to be the predominant
nitrogen source for phytoplankton growth in the Laptev and East Siberian seas during our
sampling period although there were some nitrate concentrations available.
The daily carbon uptake rate (110.3 ± 88.3 mg C m$^{-2}$ d$^{-1}$) and nitrogen uptake rate (37.0 ±
25.8 mg N m$^{-2}$ d$^{-1}$) in this study were somewhat comparable to the rates previously reported
in the Arctic Ocean (Cota et al., 1996; Lee and Whitledge, 2005; Lee et al., 2012). This is a
surprising result since the water column-integrated chl-*a* concentration in this study is
significantly higher (approximately five-fold) compared to the previous results. Various
indicators determining light or nutrient-limited conditions were suggested for the mismatch
between the higher chl-*a* concentration and relatively lower carbon and nitrogen uptake
rates. However, no consistent results were obtained because of some inherent problems of
POC including all suspended organic carbon in addition to phytoplankton carbon.
The annual primary production (13.2 g C m$^{-2}$) estimated in this study is somewhat
equivalent to the indirect measurements (9.6 g C m$^{-2}$) from dissolved inorganic carbon in the
East Siberian Sea (Anderson et al., 2011). However, the satellite-based estimations (101-121
g C m$^{-2}$) reported by Arrigo and Dijken (2011) were substantially higher in the Laptev and
Siberian seas. This large discrepancy between the field-measured and satellite-derived
primary productions might be caused by the overestimated chl-*a* concentration and primary





production from CDOM of degraded phytoplankton and terrestrial origin (Guéguen et al.,
2007; Matsuoka et al., 2011). More field-measured data are needed to understand the
mismatch between the chl-*a* concentration and primary production and will be valuable for
further validation of satellite-derived primary productions in the Laptev and East Siberian
seas.


**Acknowledgments**
We thank the captain and crew of the *Akademik Fedorov* for their outstanding assistance
during the cruise. This research was supported by the Korea Research Foundation (KRF)
grant funded by the Korea government (MEST; No. 2016015679). Support for T. E.
Whitledge and D. A. Stockwell was provided by NSF grant #120347.




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



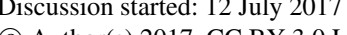



**Table caption**
Table 1. Geographical and physical information of the productivity stations in the Laptev
and East Siberian seas. Sea ice concentration was retrieved from National Snow & Ice Data
Center during the cruise period in 2013.




**Figure captions**

Figure 1. Hydrographic stations and sea ice concentration in NABOS 2013. Red dots represent productivity-measured stations. Sea ice concentration data provided from National Snow & Ice Data Center were averaged during the cruise period in 2013.

Figure 2. Spatial distribution of major inorganic nutrient concentrations (mmol m$^{-2}$) integrated from surface to 50 m water depth in the Laptev and East Siberian seas during the cruise period in 2013. a) $NO_2+NO_3$, b) $NH_4$, c) $PO_4$, and d) $SiO_4$.

Figure 3. Spatial distribution of chl-$a$ concentration (mmol m$^{-2}$) integrated from surface to 50 m water depth in the Laptev and East Siberian seas during the cruise period in 2013.

Figure 4. Compositions of size-fractionated chl-$a$ concentration (mmol m$^{-2}$) integrated from surface to 50 m water depth in the Laptev and East Siberian seas during the cruise period in 2013. a) 100 % light depth, b) 30 % light depth, and c) 1 % light depth.

Figure 5. Spatial distribution of hourly carbon uptake rates of phytoplankton (mg C m$^{-2}$ h$^{-1}$).

Figure 6. Spatial distribution of hourly nitrate (red) and ammonium (yellow) uptake rates of phytoplankton (mg N m$^{-2}$ h$^{-1}$).

Figure 7. Spatial distribution of $f$-ratio in the Laptev and East Siberian seas during the cruise period in 2013.




Table 1. Geographical and physical information of the productivity stations in the Laptev and East Siberian seas. Sea ice concentration was retrieved from National Snow & Ice Data Center during the cruise period in 2013.

| Station | Location | | Date (mm/dd/yyyy) | Depth (m) | Sea surface temperature (°C) | Sea surface salinity (psu) | Sea ice concentration (%) |
|---|---|---|---|---|---|---|---|
| | Longitude (°E) | Latitude (°N) | | | | | |
| AF005 | 109.2031 | 78.7811 | 8/25/2013 | 283 | -0.08 | 31.42 | 0 |
| AF006 | 118.4494 | 77.5925 | 8/26/2013 | 1244 | 0.75 | 31.36 | 0 |
| AF011 | 125.8045 | 77.4005 | 8/27/2013 | 1543 | 1.62 | 30.01 | 0 |
| AF019 | 125.7401 | 79.4156 | 8/28/2013 | 3196 | -1.6 | 32.44 | 25 |
| AF024 | 125.6861 | 80.7248 | 8/29/2013 | 3730 | -1.48 | 30.96 | 45 |
| AF036 | 141.5607 | 80.1791 | 9/1/2013 | 1480 | -1.22 | 28.29 | 25 |
| AF041 | 149.3758 | 79.8456 | 9/2/2013 | 561 | -1.57 | 29.86 | 60 |
| AF044 | 154.9831 | 80.2246 | 9/3/2013 | 1904 | -1.67 | 30.91 | 100 |
| AF049 | 137.7743 | 78.9502 | 9/5/2013 | 1552 | 1.57 | 29.09 | 0 |
| AF057 | 128.8313 | 77.9848 | 9/5/2013 | 2325 | 1.49 | 30.25 | 0 |
| AF061 | 125.825 | 78.399 | 9/6/2013 | 2700 | -0.07 | 31.39 | 10 |
| AF068 | 107.3858 | 79.7628 | 9/10/2013 | 1200 | -0.35 | 32.57 | 0 |
| AF071 | 112.0952 | 82.0163 | 9/11/2013 | 3530 | -1.73 | 31.86 | 65 |
| AF072 | 107.4838 | 81.4388 | 9/12/2013 | 3349 | -1.75 | 32.37 | 40 |
| AF080 | 102.3065 | 80.6008 | 9/13/2013 | 315 | -1.14 | 32.81 | 0 |
| AF091 | 97.5466 | 82.3014 | 9/14/2013 | 2959 | -1.32 | 33.3 | 0 |
| AF095 | 94.7876 | 83.7409 | 9/15/2013 | 3668 | -1.76 | 32.36 | 40 |
| AF100 | 90.0078 | 83.7489 | 9/16/2013 | 3410 | -1.49 | 33.29 | 0 |
| AF116 | 66.8714 | 81.3366 | 9/19/2013 | 530 | 0.47 | 33.44 | 0 |





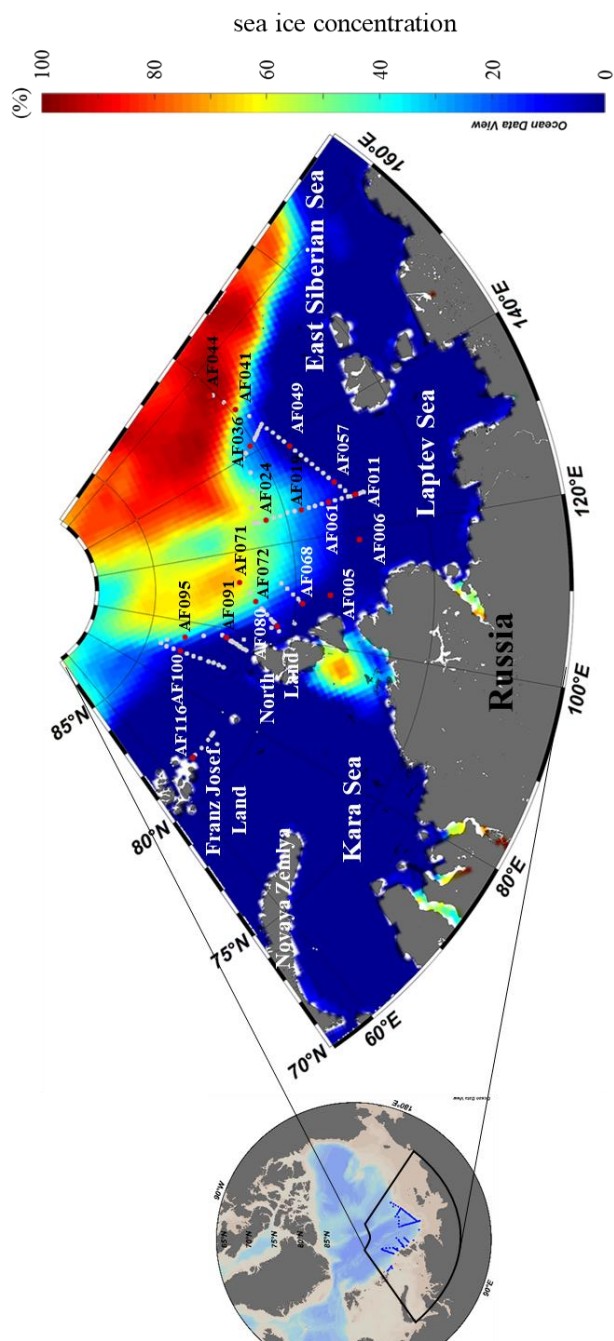

Fig. 1. Hydrographic stations and sea ice concentration in NABOS 2013. Red dots represent productivity-measured stations. Sea ice concentration data provided from National Snow & Ice Data Center were averaged during the cruise period in 2013.





Fig. 2. Spatial distribution of major inorganic nutrient concentrations (mmol m$^{-2}$) integrated from surface to 50 m water depth in the Laptev and East Siberian seas during the cruise period in 2013. a) NO2+NO3, b) NH4, c) PO4, and d) SiO4.





Fig. 3. Spatial distribution of chl-a concentration (mmol m$^{-2}$) integrated from surface to 50 m water depth in the Laptev and East Siberian seas during the cruise period in 2013.


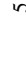

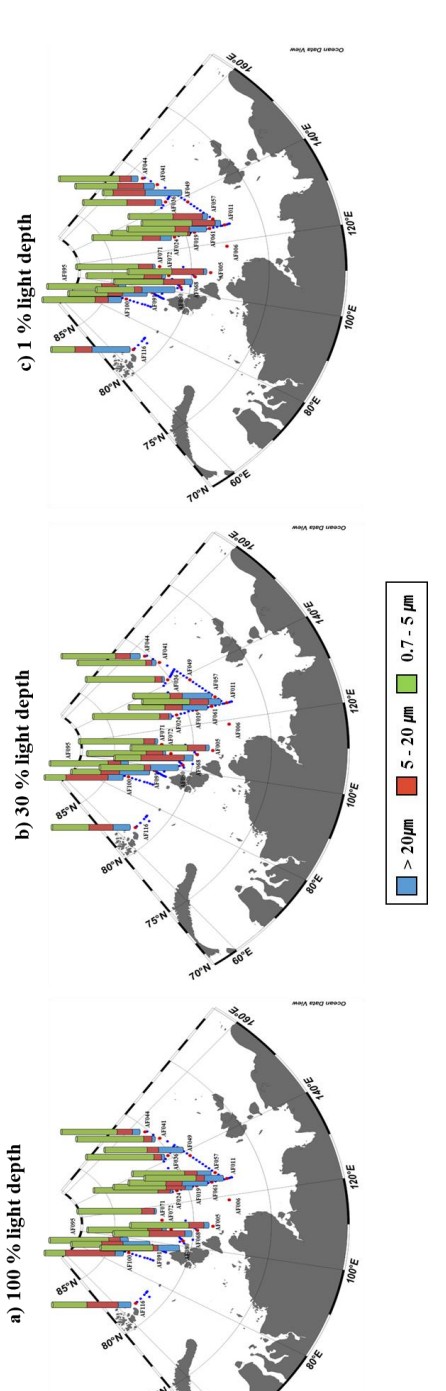

Fig. 4. Compositions of size-fractionated chl-a concentration (mmol m$^{-2}$) integrated from surface to 50 m water depth in the Laptev and East Siberian seas during the cruise period in 2013. a) 100 % light depth, b) 30 % light depth, and c) 1 % light depth.





Fig. 5. Spatial distribution of hourly carbon uptake rates of phytoplankton (mg C m$^{-2}$ h$^{-1}$).






Fig. 6. Spatial distribution of hourly nitrate (red) and ammonium (yellow) uptake rates of phytoplankton (mg N m$^{-2}$ h$^{-1}$).





Fig. 7. Spatial distribution of *f*-ratio in the Laptev and East Siberian seas during the cruise period in 2013.


