# Peer review of "Field-obtained carbon and nitrogen uptake rates of phytoplankton"

_Biogeosciences, 2017_

## Referee Comment (RC1) · Anonymous Referee #1 · 3 Aug 2017

The authors investigated the carbon and nitrogen (nitrate and ammonia) assimilation rates of phytoplankton in the Laptev and East Siberian Seas in late summer of 2013. Overall, I agree that the data obtained from this study are precious to better understand the biogeochemical and ecosystem processes of the less studied regions in the Arctic. However, in my view, the present manuscript is too descriptive, and it contains a number of ambiguous or uncertain issues. For example, below are a few severe weaknesses in this paper. As a result, I am sorry that I cannot recommend this paper for publication in the journal Biogeosiences at the present form.

1) Lack of optical data during observation. Even for the determination of optical depths,

the authors used a legacy Secchi disk technique. Please clarify the accuracy of the optical depths determined in this study. If not, the primary production data may not be reliable – an underwater PAR sensor or spectroradiometer should be used for determining the euphotic layers. In this study, the authors incubated the seawater samples for 4 to 6 hours on deck. However, no information is available for the surface PAR during incubation. Were these irradiance levels constant among stations? Also, the authors assumed 24-h daylight conditions in the summer period (L186–187). Were the light levels also constant at every station throughout the day? Please clarify these optical measurement issues. As an explanation for the lower f-ratio values observed in this study, the authors suggested potential light-limited conditions for phytoplankton growth in the study period (L220–222). Unfortunately, the authors did not show any optical or bio-optical data such as photosynthesis-irradiance parameters.

2) For a comparison between in situ and satellite remotely sensed primary production in the study area, the authors solely used the mean value in the study area during 1998–2008 reported by Arrigo and Dijken (2011) with a few assumptions. As a conclusion of this study, the authors noted that further careful validation would be required for the use of satellite data (L285–288 and L322–325). It is a shame that the authors did not make any effort to match up their in situ data with satellite-based estimates in the observation period more precisely.

Minor comments: L22: p > 0.01. Is this level statistically significant? L22: Remove "Unexpectedly" from the sentence. It could be common that the data obtained were within the previous reported values. L52: Delete "of primary producers" from the sentence. The words are redundant. L58–59: Cite a reference at least for the sentence that the Laptev and East Siberian seas are situated on the wildest and shallowest continental shelf in the world. L65–66: List the references chronologically. L72: marine ecosystems L97: Lee et al., 2007; 2012; Yun et al., 2015). L98: How did the authors convert Secchi disc depth to light intensity? L101: NaH13CO3? L108–109; Did the authors remove particulate inorganic nitrogen? If not, particulate nitrogen (PN) would

be a better expression. L115: How about the discrimination factor for 13C/12C? L129: Insert "values" between "salinity" and "ranged". L133: from the surface L139–140: I was a bit confused with the sentence that they were relatively higher in the Laptev Sea than in the East Siberian Sea. How did the authors separate the former from the latter? Where is the boundary between the two seas? Also, in Table 1, please classify the stations into the two seas. L141: the patter of silicate concentration showed opposite . . .. The verb "appear" is an intransitive verb, so it cannot be used for the passive. L148: phosphate and nitrate were so low . . .. L152: concentrations L155: Again, p > 0.01. Is this statistically significant? L169: at the surface L172: rates were L266: these production levels L267: mean production estimates L344: The "2" in CO2 should be subscript. L381: The "13" should also be subscript. L427: at the productivity measurement stations L452: Use subscript for the number of NO2+NO3, NH4, PO4, and SiO4. L439, 441, 453, and 454: The unit of chl-a concentration would be mg m-2. Fig. 4: Insert a space between "20" and "$\mu$m".

---

## Referee Comment (RC2) · Anonymous Referee #2 · 17 Aug 2017

The manuscript " Field-obtained carbon and nitrogen uptake rates of phytoplankton in the Laptev and East Siberian seas" by Lee et al.  compared their measurements of nitrate, chl-a, and primary production rates in the Laptev and E. Siberian Seas with those from other studies.  Due to short period of observations at each location and time, the comparisons are still quite rough for annual primary production estimate, but do provide a reference for these understudied regions compared to other parts of the Arctic Ocean.  I recommend publish it with some revisions. One suggestion is to add more discussion and figures showing some vertical profiles of the nitrate and chl-a concentration in the upper (100~50%) and lower (50~1%) photic depth, because there are large differences between them as described in the paper. The vertical differences

of nutrient and chl-a might be another factor that affected the spatial differences of primary production among stations. Other minor revisions: Line 16 'Field-measured ' change to 'Field measurements of '. Add the year and month of the measurement in this sentence. Line 78 remove 'Field-measured'

---

## Author Comment (AC1) · 4 Sep 2017

The authors investigated the carbon and nitrogen (nitrate and ammonia) assimilation rates of phytoplankton in the Laptev and East Siberian Seas in late summer of 2013. Overall, I agree that the data obtained from this study are precious to better understand the biogeochemical and ecosystem processes of the less studied regions in the

[Figure]

Arctic. However, in my view, the present manuscript is too descriptive, and it contains a number of ambiguous or uncertain issues. For example, below are a few severe weaknesses in this paper. As a result, I am sorry that I cannot recommend this paper for publication in the journal Biogeosiences at the present form. =>We carefully revised our manuscript based on the comments as below.

1) Lack of optical data during observation. Even for the determination of optical depths, the authors used a legacy Secchi disk technique. Please clarify the accuracy of the optical depths determined in this study. If not, the primary production data may not be reliable – an underwater PAR sensor or spectroradiometer should be used for determining the euphotic layers. => It would be better to have radiance or optical measurements for more accurate estimation of euphotic depths or diffuse attenuation coefficients for PAR, Kd(PAR). Since we, however, have no underwater PAR sensor (and/or optical instruments) available due to logistic problems (we missed our luggage from airplanes on the way to the Arctic cruise and received them two months later after the cruise), the light depths were determined by Secchi disc which has been widely and commonly used in various oceans as well as the Arctic Ocean to derive euphotic depth and Kd(PAR) (Son et al., 2005; Tremblay et al., 2000; Lee et al. 2012; Lee et al., 2017a; Lee et al., 2017b). From several previous studies in the Arctic Ocean, we are pretty much confident with the Secchi depth to get the euphotic depth since the comparison of the light depths between the two methods of Secchi disc and underwater PAR sensor were matched quite well. We added this sentence in line 102-111, pages 6-7. In this study, the authors incubated the seawater samples for 4 to 6 hours on deck. However, no information is available for the surface PAR during incubation. Were these irradiance levels constant among stations? Also, the authors assumed 24-h daylight conditions in the summer period (L186–187). Were the light levels also constant at every station throughout the day? Please clarify these optical measurement issues. =>We incubated the seawater samples on deck under natural light conditions with cooled with surface seawater for 4 to 6 hours. So, the irradiance levels were not constant among stations like natural light conditions (as we mentioned in line 120-123, pages 7-8). The 24-h daylight assumption for the daily carbon and nitrogen uptake rates was applied to this study for a comparison purpose with previous studies (Subba Rao and Platt 1984; Lee and Whitledge 2005; Lee et al., 2010) in line 221-222, page 13. Actual day lengths were about 20 hours per day during the cruise period. As an explanation for the lower f-ratio values observed in this study, the authors suggested potential light-limited conditions for phytoplankton growth in the study period (L220–222). Unfortunately, the authors did not show any optical or bio-optical data such as photosynthesis-irradiance parameters. => A potential light-limited condition is one of our hypothesis based on lower f-ratio despite with some nitrate available within the euphotic layers and no strong relationship between f-ratio and euphotic water depth-integrated concentration of nitrite+nitrate found in this study. However, the conclusion might be needed for caution as the reviewer's comment. We added more discussion on that in line 257-266, page 15. 2) For a comparison between in situ and satellite remotely sensed primary production in the study area, the authors solely used the mean value in the study area during 1998–2008 reported by Arrigo and Dijken (2011) with a few assumptions. As a conclusion of this study, the authors noted that further careful validation would be required for the use of satellite data (L285–288 and L322–325). It is a shame that the authors did not make any effort to match up their in situ data with satellite-based estimates in the observation period more precisely. => We hoped to do that, but unfortunately no annual primary production estimated in 2013 by Arrigo and Dijken (2011) makes difficult for a direct comparison of annual productions in 2013 between our measured rates and their satellite-based rates. We discussed more on potential reasons for the discrepancy between this study and satellite study in line 319-326, pages 18-19 and line 339-352, page 20. Minor comments: L22: p > 0.01. Is this level statistically significant? => We revised it in line 22. L22: Remove "Unexpectedly" from the sentence. It could be common that the data obtained were within the previous reported values. => We removed it. L52: Delete "of primary producers" from the sentence. The words are redundant. =>We deleted it. L58–59: Cite a reference at least for the sentence that the Laptev and East Siberian seas are situated on the wildest and shallowest continental shelf in the world. => We added the reference in line 59. L65–66: List the references chronologically. => We revised it in line 68-69. L72: marine ecosystems => We revised it. L97: Lee et al., 2007; 2012; Yun et al., 2015). => We revised it. L98: How did the authors convert Secchi disc depth to light intensity? => We rephrased the sentence in line 102, page 6. L101: NaH13CO3? L108–109; Did the authors remove particulate inorganic nitrogen?  If not, particulate nitrogen (PN) would be a better expression. => We revised it with PN. L115: How about the discrimination factor for 13C/12C? => Actually, we did not consider the discrimination factor for 13C/12C since it is considered too low.  We added some potential underestimated in line 135-137, page 8. L129: Insert "values" between "salinity" and "ranged". => We inserted it in line 147, page 9. L133: from the surface => We revised it. L139–140: I was a bit confused with the sentence that they were relatively higher in the Laptev Sea than in the East Siberian Sea.  How did the authors separate the former from the latter?  Where is the boundary between the two seas?  Also, in Table 1, please classify the stations into the two seas. => We rephrased the sentence (Laptev Sea=> Western part; East Siberian Sea=> Eastern part) in line 165-169, page 10, since it is not that clear for the separation. L141: the patter of silicate concentration showed opposite... The verb "appear" is an intransitive verb, so it cannot be used for the passive. => We revised it in line 167, page 10. L148: phosphate and nitrate were so low... => We revised it in line 174, page 10. L152: concentrations => We revised it in line 185, page 11. L155: Again, p > 0.01. Is this statistically significant? => We revised it in line 188, page 11. L169: at the surface => We revised it in line 203, page 12. L172: rates were => We revised it in line 206, page 12. L266: these production levels => We revised it in line 316, page 18. L267: mean production estimates => We revised it in line 317, page 18. L344: The "2" in CO2 should be subscript. => We revised it in line 413, page 24. L381: The "13" should also be subscript. => We revised it in line 458, page 25. L427: at the productivity measurement stations => We revised it in line 529, page 28. L452: Use subscript for the number of NO2+NO3, NH4, PO4, and SiO4. => We subscripted them. L439, 441, 453, and 454: The unit of chl-a concentration would be mg m-2. => We revised them all. Fig. 4: Insert a space between "20" and "_m". => We revised it.

Please also note the supplement to this comment:
https://www.biogeosciences-discuss.net/bg-2017-234/bg-2017-234-AC1-supplement.pdf

**Supplement:**

**Field-obtained carbon and nitrogen uptake rates of phytoplankton**

**in the Laptev and East Siberian seas**

**Sang Heon Lee[1*], Jang Han Lee[1], Howon Lee[1], Jae Joong Kang[1], Jae Hyung Lee[1],**

**Dabin Lee[1], SoHyun An[1], SeungHyun Son[2], Dean A. Stockwell[3], Terry E. Whitledge[3]**

* corresponding author

[1]Department of Oceanography, Pusan National University, Busan 609-735, Korea

[2]CIRA, Colorado State University, Fort Collins, CO, USA

[3]Institute of Marine Science, University of Alaska, Fairbanks, AK 99775, USA

**Abstract**

[revised manuscript text omitted]

Especially, primary production measurements are chronically scarce in this region based on the ARCSS-PP (Arctic System Science Primary Production) database 1950 to 2007 (Hill et al., 2017). Although various physical data on hydrography and ocean circulation have been reported by the continuous NABOS (Nansen and Amundsen Basins Observational System)

program (Dmitrenko et al., 2006; Bauch et al., 2014; Aksenov et al., 2011; Polyakov et al.,

2007; Aksenov et al., 2011; Bauch et al., 2014), no *in situ* measurements of recent phytoplankton productivity or nutrient concentrations have been conducted in the Laptev or

East Siberian seas during the program.

In this study, *in situ* carbon and nitrogen uptake rates of phytoplankton were measured to quantify the primary productivity and evaluate nitrogen uptake in the Laptev and East

Siberian seas as part of the NABOS program. These data will provide the basic groundwork for future monitoring of the marine ecosystems as it responds to ongoing climate change in the Laptev and East Siberian seas and will provide valuable *in situ* measurements for validating the ranges of phytoplankton primary production estimated from satellite ocean color data.

**2. Materials and Methods**

Carbon and nitrogen uptake rates of phytoplankton were measured at 19 monitoring stations selected from a total of 116 NABOS stations (Fig. 1; Table 1) in the Laptev and East Siberian seas from August 21 to September 22, 2013 onboard the

Russian vessel *"Akademik Fedorov"*. After samples for concentrations of major inorganic nutrient and chlorophyll-*a* (chl-*a*) were collected at 19 productivity stations, they were analyzed onboard during the cruise. Nutrient concentrations (nitrate, nitrite, ammonia, phosphate, and silicate) were analyzed using an Alpkem Model 300 Rapid Flow Nutrient

Analyzer (5 channels) based on the method of Whitledge et al. (1981). Total and size- fractionated chl-*a* samples were obtained from 6 light depths (100, 50, 30, 12, 5, and 1 %)

and 3 light depths (100, 30, and 1%), respectively. The chl-*a* samples were prepared based on the same procedure reported from previous studies in the Arctic Ocean (Lee et al., 2005;

Lee et al., 2012). Water samples for chl-*a* concentrations were filtered onto Whatman GF/F

(24 mm) and samples for size-fractionated chl-*a* were passed sequentially through 20 μm and 5 μm pore-sized Nucleopore filters (47 mm) and 0.7 μm pore-sized Whatman GF/F

filters (47 mm). The filters were kept frozen in a freezer (-80 °C) before further analysis.

The frozen chl-*a* samples were extracted in 90% acetone at −5°C for 24 hours, and the concentrations were measured on board using a pre-calibrated Turner Designs model 10-AU

fluorometer.

On-deck incubations for carbon and nitrogen uptake rates of phytoplankton were conducted using a $^{13}$C-$^{15}$N-dual tracer technique previously performed in various regions of the Arctic

Ocean (Lee and Whitledge 2005; Lee et al., 2007; & 2012, Yun et al., 2015). Six *in situ*

photic depths (100, 50, 30, 12, 5, and 1%) were determined at each station by converting

Secchi disc depth to light intensitydepth according to Lambert-Beer's law. It would be better to have radiance or optical measurements for more accurate estimation of euphotic depths or diffuse attenuation coefficients for PAR, $K_d$(PAR). Since we do not have underwater PAR

sensor (and/or optical instruments) due to logistic problems, the light depths were determined by Secchi disc which has been widely and commonly used in various oceans as well as the Arctic Ocean to derive euphotic depth and $K_d$(PAR) (Son et al., 2005; Tremblay et al., 2000; Lee et al. 2012; Lee et al., 2017a; Lee et al., 2017b). From several previous studies in the Arctic Ocean, we compared the light depths between the two methods of

Secchi disc and underwater PAR sensor and found that they were matched quite well (unpublished data).

Seawater samples at each light depth were transferred from the Niskin bottles to acidcleaned polycarbonate incubation bottles (approximately 1 L) matched each light depth.

Then, heavy isotope-enriched (98−99 %) solutions of $NaH^{13}CO_3$, $K^{15}NO_3$, or $^{15}NH_4Cl$ were added to the polycarbonate incubation bottles at concentrations of ~0.3 mM, ~0.8 μM, and

~0.1 μM for $^{13}CO_2$, $^{15}NO_3$, and $^{15}NH_4$, respectively. The carbon isotope enrichment was 5–

10% of the total inorganic carbon in the ambient water determined during the cruise. In contrast, the concentrations of $^{15}NO_3$ and $^{15}NH_4$ additions were greater than 10 % of the ambient nitrate and ammonium concentrations at several stations with very low concentrations. The waters injected with isotopes were incubated in big incubators on deck under natural light conditions with cooled with surface seawater for 4 to 6 hours. So, the light conditions were not constant during the incubation hours among the productivity stations. After 4 to 6 hourthe incubations on deck done, the filters used for the isotopic measurements as well as particulate organic carbon (POC) and particulate nitrogen (PONPN) were immediately preserved at –20°C for further mass spectrometric analysis (Finnigan Delta+XL) in the stable isotope laboratory of University of Alaska Fairbanks, US.

The uncertainties for $\delta^{13}$C and $\delta^{15}$N measurements were ± 0.1‰ and ± 0.3 ‰, respectively.

Calculations of the carbon and nitrogen uptake rates of phytoplankton were based on the methods from Hama et al. (1983) and Dugdale and Goering (1967). Carbon uptake rates were obtained as follows:

Carbon uptake rate = $POC_{incubation} \times [^{13}C_{excess}/ (^{13}C_{enriched} * t)]$, where $POC_{incubation}$ is the concentration of particulate organic carbon after incubation,

$^{13}C_{excess}$ is the excess $^{13}$C [concentration of $^{13}$C in the particulate phase after incubation –

natural abundance of $^{13}$C in the particulate phase], $^{13}C_{enriched}$ is the $^{13}$C enrichment in the dissolved fraction, and t is the time duration of incubation in hours. Since the discrimination factor for $^{13}$C/$^{12}$C (1.025; Hama et al., 1984) was not considered, the production rate calculated in this study could be somewhat underestimated. Nitrogen uptake rate was obtained same as carbon uptake rate. Dark carbon uptake values were subtracted from light carbon uptake values since the measured dark uptake rates were assumed from bacterial processes (Gosselin et al. 1997). Integrated values of the carbon and nitrogen uptake rates of phytoplankton were calculated from surface (100 %) to 1 % light depths based on the trapezoidal rule. The *f* ratio was calculated as a fraction of nitrate uptake rate to the sum of nitrate and ammonium uptake rates in this study (Eppley and Peterson, 1979).

**3. Results and Discussion**

The sea surface temperature and salinity values ranged from −1.76 °C to 1.62 °C and

28.29 to 33.44, respectively (Table 1). Sea ice concentration averaged during the cruise period in 2013 retrieved from National Snow & Ice Data Center ranged from 0 % to 100 %

(Table 1).

The vertical concentrations of major inorganic nutrients except nitrite+nitrate shown in

Fig. 2 were generally consistent from surface to 1 % light depth at each station although they were largely variable among the productivity stations. In comparison, the concentrations of nitrite+nitrate were homogeneous within 20 m water depth (approximately 30 % light depth)

at the most stations and then increased rapidly below the depth. The concentration of nitrite+nitrate (mostly nitrate) at surface ranged from 0 μM to 2.11 μM (average ± S.D. =

0.53 ± 0.65 μM). The concentrations of major inorganic nutrients (nitrite+nitrate, were integrated from surface to 50 m water depth because the average euphotic water column was 49.6 m (S.D. = ± 10.6 m) during our cruise period in 2013 (Fig. 3a-d). The concentrations of nitrite+nitrate and ammonium were 19.3-

189.3 mmol m$^{-2}$ and 2.5-39.7 mmol m$^{-2}$, respectively (Fig. 3a & b). The concentration of

DIN (nitrite+nitrate+ammonium) was 25.8-213.7 mmol m$^{-2}$. The concentrations of phosphate and silicate were 7.6-39.7 mmol m$^{-2}$ and 19.5-329.7 mmol m$^{-2}$, respectively (Fig.

3c & d). Generally, high concentrations of nitrite+nitrate and phosphate were found at

AF005, AF068, AF071, and AF100 and they were relatively higher in the western part than in the eastern part of the studied region (Fig. 3a &

c). In contrast, the pattern of silicate concentration  showed opposite as those of nitrite+nitrate and phosphate. The silicate concentration was higher in the eastern side than in the western part (Fig. 2d). Generally, the integrated nutrient concentrations were not depleted within the euphotic depths. Rather, they (except silicate) were nearly depleted in the upper layers (< 10 m), which represented approximately

50 % light depth in this study (Fig. 2). Our findings are consistent with the previous results in the Laptev and East Siberian seas obtained by Codispoti and Richards (1968) who observed that the concentrations of phosphate and nitrate were so low as to indicate nutrient limitation for phytoplankton production in the upper layers.

The vertical patterns of chl-*a* concentrations were largely variable from surface to

1 % light depth among the productivity stations but the chl-*a* concentrations generally decreased with depth except several stations with strong sub-surface chl-*a* maximum layers (Fig. 4). Surface chl-*a* concentrations ranged from 0.23 mg chl-*a* m$^{-3}$ at AF049 to 2.05 mg chl-*a* m$^{-3}$ at AF019 with an average of 0.64 mg chl-*a* m$^{-3}$ (S.D. = ± 0.51 mg chl-*a* m$^{-3}$) which are significantly (t-test, $p < 0.01$) higher than those (< 0.1 mg chl-*a* m$^{-3}$) in previous studies (Lee and Whitledge, 2005; Lee et al., 2012; Yun et al., 2015) from different regions in the

Western Arctic Ocean. Water column-integrated chl-*a* concentrations from surface to

[revised manuscript text omitted]

This provides indirect evidence for potential light-limited conditions for phytoplankton growth in the Laptev and East Siberian seas during the study period. However, we need to be cautious for the conclusion of light-limited phytoplankton growth in this study. Based on the size-fractionated chl-$a$ concentrations, major contributors to the phytoplankton community in this region were small sized-cells throughout the euphotic water columns in this study (Fig. 6). Generally, these small sized-cells prefer ammonium than nitrate as a nitrogen source for their growth (Tremblay et al., 2002; Lee et al., 2012). In fact, Lee et al.

(2012) observed significantly lower $f$-ratios for small sized-cells in the Western Arctic

Ocean. Therefore, the low $f$-ratios in this study could be caused by relative ammonium preference of the small cells-dominant phytoplankton community in in the Laptev and East

Siberian seas during the study period.

Codispoti et al. (2013) suggests that a nutrient-limited condition of phytoplankton production exists in the Laptev and East Siberian seas because of limited inorganic nutrient availability (phosphate and nitrate) at the surface (Codispoti and Richards, 1968). Generally, carbon/nitrogen (C/N) ratios have been used to an indicator of nutrient condition of phytoplankton (Smith and Sakshaug 1990; Lee and Whitledge, 2005; Yun et al., 2012). For example, high C/N ratios are often indicative of nitrogen deficiency for phytoplankton growth (Smith and Sakshaug 1990). The C/N ratio of particulate organic matters and assimilated C/N ratio averaged from all the productivity stations were 7.23 (± 5.51) and 7.03

(± 5.14), respectively in this study. These C/N ratios similar to the Redfield ratio (6.6)

indicate no strong nutrient-limited condition of phytoplankton production during the cruise period in 2013. In this study, the relatively lower daily carbon uptake rate despite of significantly higher chl-$a$ concentration could be caused by a potential light-limited growth condition of phytoplankton in the Laptev and East Siberian seas. In fact, there was no strong relationship between water column-integrated POC and chl-$a$ concentrations as representative biomass for phytoplankton and daily carbon uptake rate in this study ($R^2$ =

0.001), which indicates that phytoplankton biomass itself is not a main factor for the different carbon uptake rates from the productivity stations during this cruise period.

However, no relationship between POC and chl-*a* concentrations in this study could be caused by natural characteristics of POC samples since it normally includes all suspended organic carbon (detritus, bacteria, microzooplankton, etc.) as well as phytoplankton carbon.

In general, the ratio of C/chl-*a* is lower for phytoplankton under low light conditions than under high light conditions, although it is highly variable depending on all other environmental conditions that affect the growth rate of phytoplankton (Smith and Sakshaug,

1990). Based on cultures of polar or subpolar phytoplankton, the ratio ranges from 20-50 in low light conditions to 100-200 in high light conditions (Smith and Sakshaug, 1990). In fact,

Lee and Whitledge (2005) found that ratios in the Canada Basin (16.8 at approximately 2%

light level of surface irradiance to 314 at surface water) were comparable to the laboratory- measured ratios. In this study, the C/chl-*a* ratio averaged from all the productivity stations was 290.8 (S.D. = ± 164.4), which indicates no light-limited condition. Since POC contains not only phytoplankton carbon but also all suspended organic carbon, the C/chl-*a* ratio might not be a good indicator in the Laptev and East Siberian seas with large terrestrial inputs during the ice-free summer season (Macdonald et al., 2010; Anderson et al., 2011).

At this point, we do not have a good explanation for the lower carbon uptake rate despite of substantially high chl-*a* concentration of phytoplankton in the Laptev and East

Siberian seas during our observation period. Based on various indicators in this study, the growth of phytoplankton in the Laptev and East Siberian seas may have experienced a light- limited condition as well as the nutrient-limited condition, both of which are generally considered to occur in the Arctic Ocean during the summer periods (Codispoti and Richards,

1968; Codispoti et al., 2013).

For a comparison purpose, same assumptions used previously in the Arctic Ocean were applied for estimating the annual primary production in this study. Assuming Based on a 120-day growing season and the sameconstant daily productivity over the growing season in the Arctic Ocean (Subba Rao and Platt, 1984; Gosselin et al. 1997; Lee and Whitledge

2005; Lee et al. 2012; Yun et al., 2015), the annual primary production (13.2 g C m$^{-2}$) during the arctic vegetative season roughly estimated in this study is comparable to the indirect estimation (9.6 g C m$^{-2}$) from drawdown measurements of dissolved inorganic carbon in the

East Siberian Sea (Anderson et al., 2011). In addition, Hill et al. (2017) confirmed the annual primary production of 8 g C m$^{-2}$ in the East Siberian Sea which is consistent with other estimates (Codispoti et al., 2013; Popova et al., 2010), based on the seasonal *in situ*

primary production data from the ARCSS-PP database. However, these productions levels are substantially (approximately one order magnitude) lower than the mean productions estimates (101-121 g C m$^{-2}$) in the Laptev and Siberian seas for 1998-2009 estimated from satellite-based measurements (Arrigo and Dijken 2011). Unfortunately, no annual primary production estimated in 2013 by Arrigo and Dijken (2011) makes difficult for a direct comparison of annual productions in 2013 between our measured rates and their satellite-based rates. In fact, uncertainties associated with each productivity-derived method specifically versus *in situ* measurements versus satellite-based methods are complicated (Grebmeier et al. 2015). *In situ* measurements could be open to various incubation artifacts whereas satellite imagery could be biased low and high in the Arctic Ocean (Grebmeier et al. 2015). An overestimation of satellite-derived primary production in the Arctic Ocean is generally caused by an overestimation of chl-*a* concentration from massive colored dissolved organic matter (CDOM) of terrestrial origin and degraded phytoplankton (Guéguen et al., 2007; Matsuoka et al., 2011; Lewis et al., 2016). Indeed, large terrestrial inputs of dissolved and particulate organic matter are transported from substantial inputs of river runoff such as from the Lena, Indigirka, and Kolyma rivers to the shelves of the Laptev and East Siberian seas during the ice-free summer season (Macdonald et al., 2010; Anderson et al., 2011). Arrigo and Dijken (2011) also discussed a potential overestimation by the CDOM which causes some overestimation in surface chl-*a* and thus net primary production from satellite-based approaches. However, they argued that the overestimation of net primary production as high as 6.1 % is nearly equivalent to the underestimated portion (7.6 %) by missing subsurface chl-*a* maximum (SCM) layer in the Arctic Ocean. In this study, the SCM layers were also detected  common in overall productivity stations

 during the cruise period (Fig. 4).

Regional coverages could cause the difference in the annual primary productions between this and their studies. The geographic sectors defined by Arrigo and Dijken (2011)

include a large part of coastal regions which have generally high primary productivities (Arrigo et al., 2008). By comparison, our productivity stations were located in deep waters (> 200 m bottom depth; Table 1) which have relatively lower primary productivities (Arrigo et al., 2008).

In spite of the considerations for potential difference between the two methods, our filed-measured annual production is surprisingly lower compared to the satellite-derived production in the Laptev and East Siberian seas

. At this stage, our simple comparison of the primary production between this study and the satellite based-study is preliminary since our productivity measurements in this study are very limited for one time period in 2013. Based on more field measurements from different seasons and years as well as coastal regions, further careful validation between the two different methods (field measurement vs.

satellite-derived approach) is needed for a better understanding of the least biologically studied region undergoing severe and ongoing environmental changes in the Arctic Ocean.

**4. Summary and Conclusion**

Field-measured phytoplankton productivity and nutrient concentrations were obtained in the Laptev and East Siberian seas, one of the least biologically studied regions in the Arctic

Ocean (Semiletov et al., 2005; Arrigo and Dijken, 2011), during the NABOS (Nansen and

Amundsen Basins Observational System) cruise from August 21 to September 22, 2013

(Fig. 1).

During the cruise period, the nutrient concentrations within the euphotic depths were not depleted although they were depleted in the upper layers which are consistent with the previous results (Fig. 2). The euphotic water column-integrated chl-$a$ concentration ($25.7 \pm$

$14.2$ mg chl-$a$ m$^{-2}$; Fig. 3) was significantly higher in this study than those previously reported in the other parts of the Arctic Ocean (Lee and Whitledge, 2005; Lee et al., 2012).

Among the different cell sizes of phytoplankton, small phytoplankton were dominant (approximately 60 %) in the Laptev and East Siberian seas (Fig. 4). Based on the low $f$-ratio ($0.28 \pm 0.17$; Fig. 7) observed in this study, ammonium appears to be the predominant nitrogen source for phytoplankton growth in the Laptev and East Siberian seas during our sampling period although there were some nitrate concentrations available within the euphotic depths.

The daily carbon uptake rate ($110.3 \pm 88.3$ mg C m$^{-2}$ d$^{-1}$) and nitrogen uptake rate ($37.0 \pm$

25.8 mg N m$^{-2}$ d$^{-1}$) in this study were somewhat comparable to the rates previously reported in the Arctic Ocean (Cota et al., 1996; Lee and Whitledge, 2005; Lee et al., 2012). This is a surprising result since the water column-integrated chl-*a* concentration in this study is significantly higher (approximately five-fold) compared to the previous results. Various indicators determining light or nutrient-limited conditions were suggested for the mismatch between the higher chl-*a* concentration and relatively lower carbon and nitrogen uptake rates. However, no consistent results were obtained because of some inherent problems of

POC including all suspended organic carbon in addition to phytoplankton carbon.

The annual primary production (13.2 g C m$^{-2}$) estimated in this study is somewhat equivalent to the indirect measurements (9.6 g C m$^{-2}$) from dissolved inorganic carbon (Anderson et al., 2011) and 8 g C m$^{-2}$ based on the ARCSS-PP database (Hill et al., 2017) in the East Siberian Sea. However, the satellite-based estimations (101-

121 g C m$^{-2}$) reported by Arrigo and Dijken (2011) were substantially higher in the Laptev and Siberian seas. This large discrepancy between the field-measured and satellite-derived primary productions

was discussed. but our simple comparison is preliminary at this stage. More field-measured data are needed to understand the mismatch between the chl-*a* concentration and primary production and will be valuable for further validation of satellite-derived primary productions in the Laptev and East Siberian seas.

**Acknowledgments**

We thank the captain and crew of the *Akademik Fedorov* for their outstanding assistance during the cruise. This research was supported by the Korea Research Foundation (KRF)

grant funded by the Korea government (MEST; No. 2016015679). Support for T. E.

Whitledge and D. A. Stockwell was provided by NSF grant #120347.

**Table caption**

Table 1. Geographical and physical information at the productivity measurement stations in the Laptev and East Siberian seas. Sea ice concentration was retrieved from National

Snow & Ice Data Center during the cruise period in 2013.

**Figure captions**

Figure 1. Hydrographic stations and sea ice concentration in NABOS 2013. Red dots represent productivity-measured stations. Sea ice concentration data provided from National Snow & Ice Data Center were averaged during the cruise period in 2013.

Figure 2. Vertical distribution of major inorganic nutrient concentrations (µM) from surface to 1 % light depth at the productivity stations in the Laptev and East Siberian seas during the cruise period in 2013. a) $NO_2+NO_3$, b) $NH_4$, c) $PO_4$, and d) $SiO_4$.

Figure 3. Spatial distribution of major inorganic nutrient concentrations (mmol m$^{-2}$) integrated from surface to 50 m water depth in the Laptev and East Siberian seas during the cruise period in 2013. a) $NO_2+NO_3$, b) $NH_4$, c) $PO_4$, and d) $SiO_4$.

Figure 4. Vertical distribution of chl-$a$ concentration (mg chl-$a$ m$^{-3}$) from surface to 1 % light depth at the productivity stations in the Laptev and East Siberian seas during the cruise period in 2013.

Figure 5. Spatial distribution of chl-$a$ concentration (mg chl-$a$ m$^{-2}$) integrated from surface to 50 m water depth in the Laptev and East Siberian seas during the cruise period in 2013.

Figure 6. Compositions (%) of size-fractionated chl-$a$ concentration  integrated from surface to 50 m water depth in the Laptev and East Siberian seas during the cruise period in 2013. a) 100 % light depth, b) 30 % light depth, and c) 1 % light depth.

Figure 57. Spatial distribution of hourly carbon uptake rates of phytoplankton (mg C m$^{-2}$ h$^{-1}$)

in the Laptev and East Siberian seas during the cruise period in 2013.

Figure 68. Spatial distribution of hourly nitrate (red) and ammonium (yellow) uptake rates of phytoplankton (mg N m$^{-2}$ h$^{-1}$) in the Laptev and East Siberian seas during the cruise period in 2013.

Figure 79. Spatial distribution of *f*-ratio in the Laptev and East Siberian seas during the cruise period in 2013.

Table 1. Geographical and physical information of the productivity stations in the Laptev and East Siberian seas. Sea ice
concentration was retrieved from National Snow & Ice Data Center during the cruise period in 2013.

| Station | Location | | Date (mm/dd/yyyy) | Depth (m) | Sea surface temperature (℃) | Sea surface salinity (psu) | Sea ice concentration (%) |
|---|---|---|---|---|---|---|---|
| | Longitude (°E) | Latitude (°N) | | | | | |
| AF005 | 109.2031 | 78.7811 | 8/25/2013 | 283 | -0.08 | 31.42 | 0 |
| AF006 | 118.4494 | 77.5925 | 8/26/2013 | 1244 | 0.75 | 31.36 | 0 |
| AF011 | 125.8045 | 77.4005 | 8/27/2013 | 1543 | 1.62 | 30.01 | 0 |
| AF019 | 125.7401 | 79.4156 | 8/28/2013 | 3196 | -1.6 | 32.44 | 25 |
| AF024 | 125.6861 | 80.7248 | 8/29/2013 | 3730 | -1.48 | 30.96 | 45 |
| AF036 | 141.5607 | 80.1791 | 9/1/2013 | 1480 | -1.22 | 28.29 | 25 |
| AF041 | 149.3758 | 79.8456 | 9/2/2013 | 561 | -1.57 | 29.86 | 60 |
| AF044 | 154.9831 | 80.2246 | 9/3/2013 | 1904 | -1.67 | 30.91 | 100 |
| AF049 | 137.7743 | 78.9502 | 9/5/2013 | 1552 | 1.57 | 29.09 | 0 |
| AF057 | 128.8313 | 77.9848 | 9/5/2013 | 2325 | 1.49 | 30.25 | 0 |
| AF061 | 125.825 | 78.399 | 9/6/2013 | 2700 | -0.07 | 31.39 | 10 |
| AF068 | 107.3858 | 79.7628 | 9/10/2013 | 1200 | -0.35 | 32.57 | 0 |
| AF071 | 112.0952 | 82.0163 | 9/11/2013 | 3530 | -1.73 | 31.86 | 65 |
| AF072 | 107.4838 | 81.4388 | 9/12/2013 | 3349 | -1.75 | 32.37 | 40 |
| AF080 | 102.3065 | 80.6008 | 9/13/2013 | 315 | -1.14 | 32.81 | 0 |
| AF091 | 97.5466 | 82.3014 | 9/14/2013 | 2959 | -1.32 | 33.3 | 0 |
| AF095 | 94.7876 | 83.7409 | 9/15/2013 | 3668 | -1.76 | 32.36 | 40 |
| AF100 | 90.0078 | 83.7489 | 9/16/2013 | 3410 | -1.49 | 33.29 | 0 |
| AF116 | 66.8714 | 81.3366 | 9/19/2013 | 530 | 0.47 | 33.44 | 0 |

[Figure]

Fig. 1. Hydrographic stations and sea ice concentration in NABOS 2013. Red dots represent productivity-measured stations. Sea ice concentration data provided from National Snow & Ice Data Center were averaged during the cruise period in 2013.

[Figure]

Fig. 2̶2. Vertical distribution of major inorganic nutrient concentrations (μM) from surface to 1 % light depth at the productivity stations in the Laptev and East Siberian seas during the cruise period in 2013. a) $NO_2+NO_3$, b) $NH_4$, c) $PO_4$, and d) $SiO_4$.

[Figure]

Fig. 2̶3. Spatial distribution of major inorganic nutrient concentrations (mmol m$^{-2}$) integrated from surface to 50 m water depth in the Laptev and East Siberian seas during the cruise period in 2013. a) NO$_2$+NO$_3$, b) NH$_4$, c) PO$_4$, and d) SiO$_4$.

[Figure]

Fig. 24. Vertical distribution of chl-*a* concentration (mg chl-*a* m$^{-3}$) from surface to 1 % light depth at the productivity stations in the Laptev and East Siberian seas during the cruise period in 2013. [5]

[Figure]

Fig. 35. Spatial distribution of chl-a concentration (mg chl-*a* mmol m$^{-2}$) integrated from surface to 50 m water depth in the Laptev and East Siberian seas during the cruise period in 2013.

[Figure]

Fig. 46. Compositions (%) of size-fractionated chl-a concentration (mmol m⁻²) integrated from surface to 50 m water depth in the Laptev and East Siberian seas during the cruise period in 2013. a) 100 % light depth, b) 30 % light depth, and c) 1 % light depth.

[Figure]

Fig. 7. Spatial distribution of hourly carbon uptake rates of phytoplankton (mg C m$^{-2}$ h$^{-1}$) in the Laptev and East Siberian seas during the cruise period in 2013.

[Figure]

Fig. 8. Spatial distribution of hourly nitrate (red) and ammonium (yellow) uptake rates of phytoplankton (mg N m$^{-2}$ h$^{-1}$) in the Laptev and East Siberian seas during the cruise period in 2013.

[Figure]

Fig. 9. Spatial distribution of *f*-ratio in the Laptev and East Siberian seas during the cruise period in 2013.